# Hypotheses for the Reasons behind Beer Consumer’s Willingness to Purchase Beer: An Expanded Theory from a Planned Behavior Perspective

**DOI:** 10.3390/foods9121842

**Published:** 2020-12-10

**Authors:** Edward Shih-Tse Wang

**Affiliations:** Graduate Institute of Bio-Industry Management, National Chung Hsing University, 250, Kuo Kuang Rd., Taichung 402, Taiwan; shihtse.wang@msa.hinet.net

**Keywords:** theory of planned behavior, social norms, perceived behavioral control, attitude, alcohol identity, beer consumption

## Abstract

Because beer is one of the most common alcoholic beverages consumed in the world, this research adopted an expanded theory of planned behavior (TPB) perspective to understand why beer consumers purchase beer. This study investigated the effects of injunctive norms, descriptive norms, attitude, and perceived behavioral control on alcohol identity and purchase intention. The possible mediating role of alcohol identity was also investigated. This study was conducted in Taiwan, and a total of 452 beer consumers participated in the survey. Structural equation modeling was used to examine the relationship among the study variables. The results revealed that alcohol identity positively influences purchase intention, and attitude positively affects alcohol identity and purchase intention. In addition, injunctive norms have a positive influence on alcohol identity, and descriptive norms positively affect purchase intention. In particular, perceived behavioral control has a negative influence on alcohol identity but has a positive influence on purchase intention. This study also found that alcohol identity mediates the attitude–purchase intention relationship. By examining the consumption behavior of beer consumers from the TPB perspective, this study contributed to an understanding of beer consumption behavior.

## 1. Introduction

On many social occasions, alcohol plays a medium role in bringing people closer. Among alcoholic beverages, beer is the most consumed beverage and plays an important role in social activities [1]. Because of the importance of the beer market, previous studies have explored beer consumer behavior from multiple perspectives. For instance, researchers have discussed this topic from the perspective of the product and brand. Researchers have found that brand equity (i.e., perceived quality, brand awareness, brand association, and brand loyalty) affects consumers’ willingness to pay a premium price [2,3,4]. Some researchers have indicated that brand familiarity [5] and design awareness [6] affect the consumer’s beer consumption attitude or behavioral intention. Other researchers have also investigated the topic from the perspective of consumer characteristics or behavior, such as lifestyle [7], ethnocentrism [8], and the demand for alcohol [9], and have found that these factors affect their beer consumption behavior.

The theory of planned behavior (TPB) proposed by Ajzen [10] is considered to be one of the most effective theories for predicting human behavior [11]. According to TPB, subjective norms, attitudes, and perceived behavioral control affect behavioral intention and behavior [12,13]. TPB has been adopted to understand food choice behavior [14] and food waste behavior [15,16,17,18]. TPB has also been applied to investigate behavioral intention and behavior related to many types of foods, such as fruits and vegetables [19], organic foods [20,21], healthy food [11], fast food [13], and genetically modified foods [22].

Because social norms can be divided into injunctive and descriptive norms [23] and subjective norms in TBP are equivalent to injunctive norms [24], some scholars have extended TPB into four dimensions: injunctive norms, descriptive norms, attitude, and perceived behavioral control [25,26,27]. Injunctive norms refer to the degree of social approval or disapproval of the act [28]. Through people’s perceptions, injunctive norms determine whether a particular behavior is socially acceptable or not [29]. Descriptive norms are based on a view of what other people usually do, providing information regarding what seems to be the most appropriate behavior [30]. Descriptive norms are what most people do in a specific situation and are the motive for action resulting from observing what other people do [31]. Attitude can be interpreted as the extent to which a person has a positive or negative assessment of their behavior [32], and it is important in research on food consumption behavior [33]. Perceived behavioral control can be interpreted as a person’s ability to independently control their behavior [11], and the concept of perceived behavioral control is directly derived from self-efficacy, and it is similar to self-efficacy [34].

Researchers have investigated the influence of the four dimensions on various types of behavioral intentions, such as the podcast adoption intention [35], intention to publish in open access journals [27], purchase intention for organic cotton apparel [26], and intention of public reporting on food safety incidents [36]. Despite popular interest, few studies have adopted the expanded TPB to explain alcohol consumption behavior in general and beer consumption behavior in particular. As a result, knowledge on whether injunctive norms, descriptive norms, attitude, and perceived behavioral control are related to beer purchase intention is limited.

Previous studies have shown that self-identity has a crucial effect on behavior and behavioral intention [37,38]. As self-identity is “how individuals see themselves” [39], self-identity is equal to alcohol identity in the context of beer consumption, which refers to the extent to which a person considers himself or herself a “drinker.” Previous studies have integrated the expanded TPB and self-identity to investigate the direct influence of injunctive norms, descriptive norms, attitude, perceived behavioral control, and self-identity on purchase intention [37,40,41,42]. Although a previous study pointed out that attitude and perceived behavioral control positively affect self-identity [43], no study has explored the possible mediating role of self-identity in the effects of injunctive norms, descriptive norms, attitude, and perceived behavioral control on purchase intention. Thus, the following research questions are proposed:

RQ1: Do injunctive norms, descriptive norms, attitude, and perceived behavioral control affect the beer consumer’s purchase intention?

RQ 2: Are the effects of injunctive norms, descriptive norms, attitude, and perceived behavioral control on beer consumers’ purchase intention mediated by alcohol identity?

In summary, to provide beer marketers and researchers with an understanding of beer consumption behavior through an expanded TPB perspective, in this study the relationships among injunctive norms, descriptive norms, attitude, perceived behavioral control, alcohol identity, and purchase intention were explored in the context of beer consumption.

## 2. Materials and Methods

### 2.1. Hypothesis and Research Model

The study investigated how injunctive norms, descriptive norms, attitudes, and perceived behavioral control affect alcohol identity and purchase intention. Accordingly, the following hypotheses and conceptual framework (as illustrated in Figure 1) were posited to understand the relationships among these factors:

**Hypothesis** **1 (H1):***Injunctive norms positively influence alcohol identity*.

**Hypothesis** **2 (H2):***Injunctive norms positively influence purchase intention*.

**Hypothesis** **3 (H3):***Descriptive norms positively influence alcohol identity*.

**Hypothesis** **4 (H4):***Descriptive norms positively influence purchase intention*.

**Hypothesis** **5 (H5):***Attitude positively influences alcohol identity*.

**Hypothesis** **6 (H6):***Attitude positively influences purchase intention*.

**Hypothesis** **7 (H7):***Perceived behavioral control positively influences alcohol identity*.

**Hypothesis** **8 (H8):***Perceived behavioral control positively influences purchase intention*.

**Hypothesis** **9 (H9):***Alcohol identity positively influences purchase intention*.

### 2.2. Sample and Data Collection

The questionnaire survey was conducted online from 17 January 2019 to 25 February 2019. The research was conducted in Taiwan, where drinking is very popular. Since Taiwan became a member of the World Trade Organization in 2002 and the alcohol monopoly was abolished [44], studies have focused on the potential health impact of alcoholic beverage consumption [45,46]. Little research has been conducted on beer consumption behavior in Taiwan. The research sample of this study included consumers over the age of 18 years who had consumed beer in the past. Before the questionnaire was administered, consent was obtained from consumers. By using a question to identify beer consumers (i.e., Have you ever drunk beer?), we excluded invalid samples. The population size and distribution of beer consumers in Taiwan is unknown because no official statistics are available. According to Cochran’s [47] formula, the minimum sample size is 385 for a 95% confidence interval with a 0.5 permitted margin of error. Moreover, in multivariate research, the sample size should be 20 times larger than the number of observation items in the study [48]. Because 19 items were used in the study, a sample size larger than 380 was required. The 452 valid questionnaires received at the end of the survey was a greater sample size than the requisite minimum sample size.

### 2.3. Scale Development

All scales for measuring the variables, which have been shown to possess internal reliability and convergent validity, were adopted from other studies. These scales were chosen among alternative scales in the literature because they best represented the model constructs. Constructs related to injunctive norms and descriptive norms were measured using two 3-item scales developed by Wang and Lin [49]. A 3-item scale was used to measure attitude (Maio et al., 2003) [50], and perceived behavioral control was measured using a 3-item scale modified from the study of Gao et al. [51]. A 3-item scale was adapted from the study of Barbarossa and De Pelsmacker [39] to measure alcohol identity. Finally, a 4-item scale, which was slightly modified from the study of Barber and Taylor [52], was used to measure purchase intention. The questionnaire comprised 19 questions, and all items were measured on a 7-point Likert scale, ranging from 1 (completely disagree) to 7 (completely agree). The questionnaire items in this research are listed in Table 2.

## 3. Results

### 3.1. Demographics

As shown in Table 1, among respondents, 265 (58.6%) were female and 187 (41.1%) were male. In terms of age distribution, respondents aged 20–29 years comprised the largest group (66.2%), followed by respondents aged 30–39 years (13.9%). In terms of monthly income, respondents with the monthly income of NTD 30001–40000 comprised the largest group (23.2%), followed by respondents with the monthly income of NTD 20001–30000 (16.4%). Most respondents had higher education degrees, including 66.2% with college degrees and 23.7% with master’s degrees. In terms of current occupation, 28.8% (130 respondents) of the total study sample listed “student” as their current occupation, which comprised the largest group. The second largest group in terms of occupation included service and sales workers, accounting for 21.7% of the total population.

### 3.2. Convergent Validity

Using LISREL version 8.70 and structural equation modeling (SEM), the causality relationship among the variables and the overall fit of the model were analyzed. To understand the relationship between observed variables and potential variables, confirmatory factor analysis (CFA) was adopted. CFA was used to verify the factor structure of the theoretical model and to examine the reliability and validity of the measured indicators. Convergent validity denotes the degree of correlation between the results obtained for the same construct measured using different items, which can be estimated using three test methods, namely composite reliability (CR), factor loadings (λ) of the items, and average variance extracted (AVE). The value of CR should be higher than 0.7, and the value of AVE must be higher than 0.5 at least [53]. The value of the acceptable factor loading should be above 0.5 at least [54]. This study’s CFA plot is shown in Figure 2. As shown in Table 2, the CR value ranged from 0.78 to 0.98, all of which were greater than 0.7; the value of AVE ranged from 0.56 to 0.90, all of which were greater than 0.5; and the value of factor loadings ranged from 0.51 to 0.98, all of which were greater than 0.5. In summary, the standards were met for composite reliability, average variance extracted, and factor loadings; thus, the questionnaire had convergent validity.

### 3.3. Discriminant Validity

Discriminant validity denotes the degree of difference between latent constructs and other constructs [53]. The discriminant validity test is a test of whether a construct measure is empirically unique and of whether it correlates too closely with other measures from which it should differ; an indicator of discriminant validity is obtained through comparing the square root values of the AVE of each construct and correlations with other constructs in the model [54]. When the ratio of AVE from each construct exceeds the square of the coefficient representing its correlation with other constructs, the validity of the discrimination is sufficient [53]. The AVE square root values of each variable in this study were all greater than the correlation coefficient between constructs, indicating that each variable is distinct, and that discriminant validity is sufficient (As shown in Table 3).

### 3.4. Results of Structural Model Analysis and Path Analysis

The causality relationship between the variables and the overall fit of the model were analyzed using SEM. The overall fit of the structural model of this study was elucidated in terms of the following absolute-fit measures: χ2 = 492.01, d.f. = 137, χ2/d.f. = 3.591, GFI = 0.90, AGFI = 0.86, RMSEA = 0.076. The incremental-fit measures of this study were as follows: CFI = 0.98, NFI = 0.97, NNFI = 0.97. The parsimonious-fit measures of this study were as follows: PNFI = 0.78, PGFI = 0.65. Overall, the structural model had an acceptable fit. Next, according to the path analysis results, injunctive norms positively affected alcohol identity (β =0.22, *p* < 001), and descriptive norms positively affected purchase intention (β =11, *p* < 01). Thus, H1 and H4 were supported. Attitude positively affected alcohol identity (β =0.58, *p* < 001), and attitude positively affected purchase intention (β = 0.39, *p* < 001), supporting H5 and H6. In addition, perceived behavioral control positively affected purchase intention (β =0.13, *p* < 001), supporting H8. Alcohol identity positively affected purchase intention, supporting H9 (β =0.46, *p* < 001). However, H2 and H3 were not supported: alcohol identity was not significantly affected by injunctive norms (β = −0.071, *p* > 05) or descriptive norms (β = 0.05, *p* > 05). Furthermore, H7 was not supported because the results indicated the opposite direction, where perceived behavioral control negatively affected alcohol identity (β = −0.10, *p* < 01). The path analysis results are shown in Table 4. The results reveal that the model explained 59% and 66% of the variance of alcohol identity and purchase intention, respectively.

### 3.5. Results of Mediating Analysis

This study also examined the mediating role of alcohol identity in the expanded TPB. To determine whether alcohol identity has a mediating effect, the current study used the analytic method, which involves 4 steps, suggested by Baron and Kenny [55]. As shown in Table 5, alcohol identity partially mediated the effects of attitude on purchase intention, whereas alcohol identity did not have a mediation effect on the effects of injunctive norms, descriptive norms, and perceived behavioral control on repurchase intention.

## 4. Discussion

### 4.1. Theoretical and Practical Implications

This study adopted the expanded TPB to predict consumers’ willingness to repurchase beer. More specifically, the influence of injunctive norms, descriptive norms, attitude, and perceived behavioral control on purchase intention were explored, along with the potential mediating role of alcohol identity in the aforementioned relationships. Because few studies have adopted the expanded TPB to explain beer consumption behavior, this study contributes to the relevant literature by examining beer consumption behavior from a TPB perspective. The results of empirical investigation revealed that injunctive norms have a positive effect on alcohol identity, and descriptive norms have a positive effect on purchase intention. Attitude positively affects alcohol identity and purchase intention. Perceived behavioral control has a positive effect on purchase intention, but it has a negative effect on alcohol identity. Alcohol identity has a positive and significant effect on purchase intention. However, injunctive norms have no significant effect on alcohol identity, and descriptive norms do not significantly affect alcohol identity.

Purchase intention was affected by attitude [56], perceived behavioral control [57], and identification [37], consistent with other studies’ findings. However, contrary to this study’s findings, Wauters et al. [43] reported that perceived behavioral control positively (rather than negatively) affects self-identity. This inconsistency is primarily attributable to the characteristics of beer, excessive consumption of which is unhealthy and whose drinkers (even moderate ones) are reluctant to think of themselves as typical beer consumers. In particular, this study used alcohol identity as a mediating variable to explore the relationship among injunctive norms, descriptive norms, attitude, perceived behavioral control, and purchase intention. According to the results of empirical research, alcohol identity partially mediates the attitude–purchase intention relationship but does not mediate the influence of injunctive norms, descriptive norms, and perceived behavioral control on purchase intention.

This study revealed that attitude influences purchase intention. When consumers think that drinking beer is a favorable activity, they will consider buying beer products. Alcohol identity positively affects purchase intention; that is, because a person thinks he or she is a typical drinker, he or she will be more willing to purchase beer. In particular, the study results showed that descriptive and injunctive norms have different effects on drinking perceptions. Injunctive norms have a positive influence on alcohol identity but do not affect purchase intention, whereas descriptive norms affect purchase intention but do not affect alcohol identity. When a person perceives drinking beer is socially acceptable, the person will be more likely to think he or she is a typical drinker. By contrast, when a person perceives that many others drink beer, the person will be more willing to buy beer products. An interesting finding is that perceived behavioral control has a positive effect on purchase intention but a negative effect on alcohol identity. The more consumers think that they can control how much they drink, the less they think of themselves as a typical drinker, the more willing they are to consider buying beer products.

### 4.2. Limitations and Future Research Directions

First, the questionnaires were administered online, which may result in the recruitment of a biased sample population. In follow-up studies, researchers can conduct more extensive, nationwide sampling to improve the generalizability of the findings. Second, in this study, social norms were divided into injunctive norms and descriptive norms. Fang et al. [58] considered that social norms can be divided into four dimensions: injunctive norms, descriptive norms, subjective norms, and personal norms. In follow-up studies, researchers can use other classifications of social norms to discuss the research topic.

## Figures and Tables

**Figure 1 foods-09-01842-f001:**
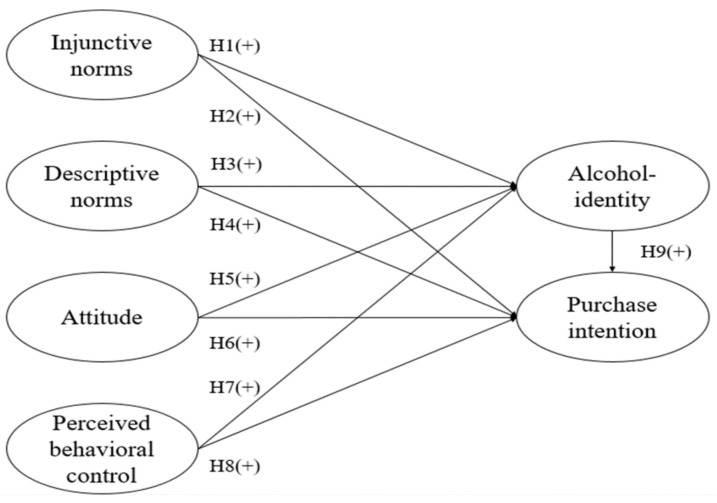
Conceptual framework.

**Figure 2 foods-09-01842-f002:**
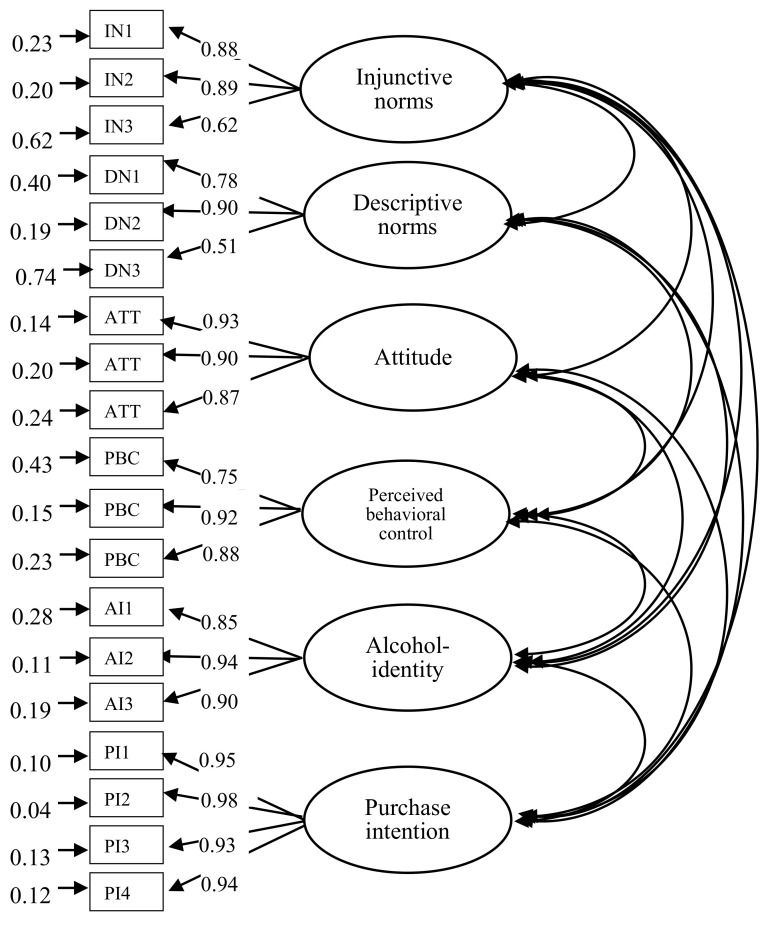
Confirmatory Factor Analysis (CFA) plot.

**Table 1 foods-09-01842-t001:** Descriptive statistical analysis.

	Descriptive Statistics	Freq.	%		Descriptive Statistics	Freq	%
Gender	Male	187	41.1%	Level of Education	Elementary school	1	0.2%
	Female	265	58.6%		Junior high school	3	0.7%
Age	Under 20	11	2.4%		Senior / vocational high school	42	9.3%
	20–29	299	66.2%		College	299	66.2%
	30–39	63	13.9%		Master’s or above	107	23.7%
	40–49	39	8.6%	Occupation	Legislators, Senior Officials and Managers	10	2.2%
	50–59	37	8.2%		Professionals	89	19.7%
	60–69	3	0.7%		Technicians and Associate Professionals	25	5.5%
Monthly Income (NTD)	Unpaid Income	73	16.2%		Clerical Support Workers	24	5.3%
	Below 20000	71	15.7%		Service and Sales Workers	98	21.7%
	20001–30000	74	16.4%		Skilled Agricultural, Forestry, and Fishery Workers	6	1.3%
	30001–40000	105	23.2%		Craft and Related Trades Workers	14	3.1%
	40001~50000	66	14.6%		Plant and Machine Operators, and Assemblers	11	2.4%
	50001~60000	29	6.4%		Elementary Laborers	5	1.1%
	60001~70000	15	3.3%		Students	130	28.8%
	Above 70000	19	4.2%		Military, police, teachers, and government employees	27	6.0%
					Others	13	2.9%

**Table 2 foods-09-01842-t002:** Convergent validity test results.

Variables	Observed Variables	λ	CR	AVE
Injunctive norms	Most people who are important to me believe that I should drink.	0.88	0.85	0.65
	Some people who have an influence on me believe that I should drink.	0.89		
	My close friends and family believe that drinking is a good habit.	0.62		
Descriptive norms	As far as I know, many people drink beer.	0.78	0.78	0.56
	Many of my friends drink beer.	0.90		
	Many of my family members drink beer.	0.51		
Attitude	I personally think that drinking is a good thing.	0.93	0.93	0.81
	I personally think that drinking is a favorable thing.	0.90		
	I personally think that drinking is a valuable thing.	0.87		
Perceived behavioral control	It’s easy for me to control how much alcohol I drink.	0.75	0.89	0.73
	I have the ability to control how much alcohol I drink.	0.92		
	I can control how much alcohol I drink without too much effort.	0.88		
Alcohol-identity	I believe that I am a person who cares about beer products.	0.85	0.93	0.81
	I believe that I am a typical beer consumer.	0.94		
	Buying beer products make me feel like a typical beer consumer.	0.90		
Purchase Intention	I plan to keep buying beer products in the future.	0.95	0.97	0.90
	I think I will continue to buy beer products in the future.	0.98		
	I’ve been in trouble at work due to my drinking habit.	0.93		
	I am still interested in buying beer products in the future.	0.94		

**Table 3 foods-09-01842-t003:** Correlations among the latent variables.

Variables	Avg.	S.D.	IN	DN	ATT	PBC	AI	PI
Injunctive norms (IN)	3.52	1.33	0.80					
Descriptive norms (DN)	5.17	1.18	0.44	0.75				
Attitude (ATT)	4.16	1.43	0.61	0.52	0.90			
Perceived behavioral control (PBC)	5.86	1.18	0.06	0.07	0.09	0.85		
Alcohol-identity (AI)	3.41	1.61	0.56	0.43	0.69	–0.06	0.90	
Purchase intention (PI)	4.83	1.57	0.47	0.48	0.70	0.07	0.72	0.95

Note: The grey spots are the squared average variance extracted values of each construct, whereas the rest of the values are correlation coefficients between constructs.

**Table 4 foods-09-01842-t004:** Research findings.

Path	Path Coefficients	T-value
H1 Injunctive norms →Alcohol-identity	0.22	4.36 ***
H2 Injunctive norms →Purchase intention	–0.07	–1.63 ^n.s^
H3 Descriptive norms →Alcohol-identity	0.05	1.03 ^n.s^
H4 Descriptive norms →Purchase intention	0.11	2.89 **
H5 Attitude →Alcohol-identity	0.58	10.05 ***
H6 Attitude → Purchase intention	0.39	6.85 ***
H7 Perceived behavioral control →Alcohol-identity	–0.10	–2.75 **
H8 Perceived behavioral control →Purchase intention	0.09	2.91 **
H9 Alcohol-identity →Purchase intention	0.46	8.66 ***

Notes: ^n.s^: not significant (*p* > 0.05). ** *p* < 0.01. *** *p* < 0.001.

**Table 5 foods-09-01842-t005:** Testing of the mediating effect (BK approach).

Mediation Path			Path Coefficients					Mediating Effect
IV	M	DV	(1) IV→M	(2) M→DV	(3) IV→DV	(4) IV+M→DV		
						M→DV	IV→DV	
IN	AI	PI	0.22 ***	0.75 ***	0.04 ^n.s^	0.47 ***	−0.07 ^n.s^	No
DN	AI	PI	0.05 ^n.s^		0.14 **		0.11 **	No
ATTI	AI	PI	0.58 ***		0.66 ***		0.38 ***	Partial
PBC	AI	PI	−0.10 *		0.04 ^n.s^		0.09 **	No

Note: IV: independent variable; M: mediator; DV: dependent variable; IN = injunctive norms; DN = descriptive norms; ATTI = attitude; PBC = perceived behavioral control; AI = alcohol identity; PI = purchase intention. ^n.s^: not significant (*p* > 0.05) * *p* < 0.05. ** *p* < 0.01. *** *p* < 0.001.

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
