# Peer review of "Hypotheses for the Reasons behind Beer Consumer’s Willingness to Purchase Beer: An Expanded Theory from a Planned Behavior Perspective"

_foods, 2020, doi:10.3390/foods9121842_

Round 1
Reviewer 1 Report
The title should be better changed in : Hypotheses why.....
The 452 valid questionnaires were selected by chance, I mean there was no a previous experiment on the amount of subjects to be involved. For the same reason the segments founds were casually identified and not planned in advance.
Table 1 is the result of what above mentioned: a casual (not random) sample recruited and not planned from the beginning of the work.
Line 183-185 seem too much pretentious, since many of the results founded look self evident.
For example Page 2 line 51-53: if the reference universe (...observing what other people do..) should be made by criminals would the AA justify the same sentence?
line 80 : which would be the information provided for beer makers?
line 89 and following: there is no trace of the Taiwan study. 452 received questionnaires of how many?
Where is the CFA plot?
Line 139 - 145 this section needs more depth.
line 158-165 self evident
line 210-211 : the same happens with drugs and narcotics
Less references are enough
Author Response
Responses to the First Reviewer’s Comments
The authors acknowledge and appreciate all these issues raised by the reviewers. Following reviewers’ suggestion and comments, the authors have modified this manuscript thoroughly. All the modified parts are indicated in blue in the manuscript.
Comments:
Q1-1. The title should be better changed in : Hypotheses why.....
A1-1. As suggested, the title has been changed to “Hypotheses for the reasons behind beer consumers’ willingness to purchase beer: an expanded theory from a planned behavior perspective.”
Q1-2. The 452 valid questionnaires were selected by chance, I mean there was no a previous experiment on the amount of subjects to be involved. For the same reason the segments founds were casually identified and not planned in advance. Table 1 is the result of what above mentioned: a casual (not random) sample recruited and not planned from the beginning of the work.
A1-2. The original version indeed failed to clearly provide a clear description of the sampling procedure and method. First, an adequate sample size was computed per the recommendations of Cochran (1977) and Guenzi and Georges (2010). We have rewritten the section per the reviewer’s suggestions. Second, the population size and distribution of beer consumers in Taiwan are unknown because no official statistics are available, which prompted our decision to not use random sampling or a quota. Per your suggestion, we have stated this point in the section on research limitations and future research directions. We believe that the use of a nationwide representative sample in future studies will improve the validity of results.
Cochran, W.G. Sampling techniques. 1977. New York: John Wiley & Sons, Inc.
Guenzi, P.; Georges, L. Interpersonal trust in commercial relationships: Antecedents and consequences of customer trust in the salesperson. Eur J Mark. 2010, 44, 114-138.
Q1-3. Line 183-185 seem too much pretentious, since many of the results founded look self evident.
line 158-165 self evident
line 80 : which would be the information provided for beer makers?
Line 139 - 145 this section needs more depth.
A1-3. We apologize for our failure to fully understand the reviewer’s suggestion, which we take to be a call for greater clarity and elaboration. The authors have rewritten these sentences and sections. If this version of the manuscript is unacceptable, we are more than willing to revise it further.
Q1-4. For example Page 2 line 51-53: if the reference universe (...observing what other people do..) should be made by criminals would the AA justify the same sentence?
line 210-211 : the same happens with drugs and narcotics
A1-4. In our study, the effects of injunctive norms, descriptive norms, attitude, and perceived behavioral control on alcohol identity and willingness to purchase beer were examined. Therefore, these factors’ effects on criminal and drug-related behaviors exceed the scope of the current study. Discussion regarding other behaviors may confuse readers of our study, and we have therefore opted to omit it.
Q1-5. line 89 and following: there is no trace of the Taiwan study. 452 received questionnaires of how many?
A1-5. Research on beer consumption behavior in Taiwan is scarce, and we have cited recent studies on alcohol (instead of beer) consumption behavior in Taiwan. Please refer to line 120-124 of the revised paper.
Q1-6. Where is the CFA plot?
A1-6. Per the reviewer’s comment, a CFA plot has been added in the revised manuscript. Please refer to Figure 2.
Q1-7. Less references are enough
A1-7. We are grateful for the reviewer’s suggestion. However, we wish to retain these references for greater clarity, as support for our arguments, and to guide the readers to the relevant literature.

Reviewer 2 Report
Good work. It's an original perspective for an important topic and market like beer consumption. Very interesting the results about injunctive norms and descriptive norms.
The authors should include a Graphic about the model and not only tables, It would be easier to understand. The work in general is very good. But the graphic would explain better the model for the readers. That's the thing should be improved in the presentation.
Author Response
Responses to the Reviewer 2’s Comments
The authors acknowledge and appreciate all these issues raised by the reviewers. Following reviewers’ suggestion and comments, the authors have modified this manuscript thoroughly. All the modified parts are indicated in blue in the manuscript.
Comments:
Q2-1. Good work. It's an original perspective for an important topic and market like beer consumption. Very interesting the results about injunctive norms and descriptive norms.
A2-1. We are grateful for the positive assessment of the manuscript.
Q2-1. The authors should include a Graphic about the model and not only tables, It would be easier to understand. The work in general is very good. But the graphic would explain better the model for the readers. That's the thing should be improved in the presentation.
A2-2. This suggestion is helpful and greatly appreciated. We have drawn a figure of the survey model. Please refer to Figure 1.

Reviewer 3 Report
The author assessed the effects of injunctive norms, descriptive norms, attitude, and perceived behavioral control on alcohol identity and purchase intention. It was interesting to read this study. The paper is well constructed, clearly written, and presents interesting results. Below are some suggestions to improve the work:
- The author used a simple of 452 consumers, is it representative of the population of Taiwan? What is the margin of error? Is it 4.61%? is it acceptable? Please provide a reference to justify it.
- The model should be graphically represented (Model Conceptualization) to show the relationships between the variables.
- The sources of the used scales are provided but the author did not explain why? Please explain to us why you chose these scales rather than others.
- In the discussion, the results of this study should be compared with previous findings of other authors.
Congratulations!
Author Response
Responses to the Reviewer 3’s Comments
The authors acknowledge and appreciate all these issues raised by the reviewers. Following reviewers’ suggestion and comments, the authors have modified this manuscript thoroughly. All the modified parts are indicated in blue in the manuscript.
Comments:
Q3-1. The author assessed the effects of injunctive norms, descriptive norms, attitude, and perceived behavioral control on alcohol identity and purchase intention. It was interesting to read this study. The paper is well constructed, clearly written, and presents interesting results.
A3-1. Thank you for your positive feedback regarding our study. The following sections contain our responses to each of your points and suggestions.
Q3-2. The author used a simple of 452 consumers, is it representative of the population of Taiwan? What is the margin of error? Is it 4.61%? is it acceptable? Please provide a reference to justify it.
A3-2. The original manuscript indeed failed to clearly describe the sampling procedure and method. We have rewritten these sections. Please refer to line 126-132.
Q3-3. The model should be graphically represented (Model Conceptualization) to show the relationships between the variables.
A3-3. This suggestion is helpful and greatly appreciated. Per your suggestion, we have drawn a figure of the survey model. Please refer to Figure 1.
Q3-4. The sources of the used scales are provided but the author did not explain why? Please explain to us why you chose these scales rather than others.
A3-4. We apologize for the lack of elaboration. Per your comment, we have stated the reasons for using the scales. Please refer to line 135-137.
Q3-5. In the discussion, the results of this study should be compared with previous findings of other authors.
A3-5. Thank you for this suggestion. Accordingly, we have rewritten the discussion section to compare our results with those of other studies. Please refer to line 276-279.

Round 2
Reviewer 1 Report
I feel satisfied after the AA revision